# A Geometric Contextual Model for Identifying Unseen Metaphors

## Abstract

We present a method for metaphor identification based on an explicitly geometric approach to modelling salient lexical associations, and test it on a task of distinguishing metaphoric from literal uses of adjective-noun phrases. By contextually projecting candidate word pairs into interpretably geometric spaces, we show that it obtains state-of-the-art performance, while providing a method which is effectively *zero-shot* – able to be applied to new unseen phrases. Our dynamically context-sensitive model is inspired by theoretical insight into the situational nature of language and cognition.

## 1 Introduction

The standard approach to the computational modelling of metaphor has been grounded in theoretical work suggesting that metaphors are mappings between conceptual domains, and has accordingly dealt with the construction of representational structures that allow symbolic mappings between semantic categories. The result has been a rich literature of both symbolic and statistical approaches to metaphor identification, interpretation, and even generation. Recent approaches in the statistical vein have tended to centre on the construction of linear algebraic representations of words – words as vectors and matrices – which can be compositionally interpreted in terms of their mathematical properties. One such example for identification of metaphor is (Gutiérrez et al., 2016), who reported strong results on a new dataset using a compositional technique involving the supervised and semi-supervised learning of adjective tensors for the classification of candidate adjective-noun metaphors.

Here, we also take a distributional semantic approach, but predicated on a slightly different theoretical premise: we hold that metaphors are linguistic phenomena that emerge dynamically in the course of language use, as a semantic phenomenon at the end of a fluid spectrum, related to the *ad hoc*, situated nature of concept formation. Our work, based on a method described by McGregor et al. (2015) for concept formation, contextually generates a new subspace for every candidate metaphor, projecting word pairs into unique geometric relationships based on an analysis of their independent word-vector representations. In addition to its contextual flexibility, the strengths of our model are its simplicity and its generality: it requires only vectors for individual words, rather than representations of the potentially metaphoric phrases themselves, and is therefore suitable for the "zero-shot" setting of identifying a novel, unseen metaphor. Analysing the subspaces produced by our model by means of a logistic regression, we report F-scores that are comparable with the supervised method described by Gutiérrez et al. and superior to their semi-supervised method.

## 2 Background

As Shutova (2010) points out, the conceptual work that has had the greatest impact across various computational models of metaphor emerged from a spate of theoretical productivity in the late 1970s and early 1980s (see Ortony, 1993, for a valuable compendium). Per Shutova's analysis, two particularly influential theories of metaphor are the *selectional preference violation* model due to Wilks (1978) and the *conceptual metaphor* model of Lakoff and Johnson (1980). Wilks takes an approach that correlates with the pragmatic maxims of Grice (1975), suggesting that metaphor constitutes some sort of breach of conceptual ex-

pectation. Lakoff and Johnson, on the other hand, in work that has been much discussed by cognitive linguists and cognitive scientists, describe metaphor in terms of isomorphic mappings between broad conceptual domains, with the anatomy of these often culturally specific domains coercing the nuances of particular metaphors.

NLP researchers have generally interpreted this theoretical position to indicate that concepts are stable categories essentially denoted by linguistic symbols, and that communication is more or less about the efficacious application of these symbols in trafficking propositions about things in the world. A result of this stance has been that the computational modelling of metaphor tends to focus on the construction of static representational structures where metaphoricity is assessed in terms of conceptually transgressive compositions of these representations: the metaphor is taken to transport meaning from one domain to another in a semantically effective way (for a comprehensive overview, see Shutova, 2015). In terms of symbolic approaches, which often incorporate some sort of either hand-made or automatically encoded knowledge base, this has resulted in models involving the chaining of inferences across sequences of symbols to discover novel conceptual relationships (Lee and Barnden, 2001) and the scaling of taxonomies such as WordNet (Veale and Hao, 2008) or custom databases (Veale, 2016).

Statistical approaches have often focused on the distributional semantic paradigm, using observations of co-occurrences across large-scale corpora to build word representations which can subsequently be manipulated through linear algebraic operations (see Clark, 2015, for an overview). So, for instance, Utsumi (2011) models metaphor comprehension as a process of clustering terms that are proximate to both elements of two-word metaphors in a latent semantic analysis model (Deerwester et al., 1990). In the specific area of metaphor identification, Turney et al. (2011) present a model which uses distributional statistics to derive sets of abstract and concrete nouns, the assumption being that metaphor can be detected as a mapping between likewise abstract and concrete domains, and Hovy et al. (2013) use a combination of word-vector representations and annotated trees to identify syntactic consistencies that indicate the presence of metaphor.

**Metaphoricity detection via mappings** In recent work, Gutiérrez et al. (2016) provide a good example of this general approach, together with a dataset and task involving the classification of a fairly evenly balanced set of literal and metaphoric adjective-noun pairs, specifically designed to include adjectives which tend to occur in both metaphoric and literal compositions. We describe the dataset in more detail in Section 4. Their approach takes the Lakoff and Johnson (1980) view of metaphor as mapping, implemented via supervised learning of distinct adjective tensors for metaphoric and literal senses (given a view of adjectives as tensors and nouns as word vectors, a technique previously employed by Baroni and Zamparelli (2010) and more generally in line with current developments in compositional distributional semantics (Mitchell and Lapata, 2010; Coecke et al., 2011)).

They test two versions of this approach. Given an adjective-noun pair $\{a, n\}$, the first version learns a tensor $A_a$ for each adjective $a$ by building vectors $\overrightarrow{n}, \overrightarrow{an}$ from observed co-occurrence statistics of the noun $n$ and the combined phrase $an$, and then minimising the error between $\overrightarrow{an}$ and the product $A_a \overrightarrow{n}$. For each adjective $a$, they learn two separate tensors $A_{M(a)}, A_{L(a)}$ for literal and metaphoric senses, learned from labelled $an$ pairings. An phrase $an$ is then considered metaphoric if $\overrightarrow{an}$ is closer to the product $A_{M(a)} \overrightarrow{n}$ than to $A_{L(a)} \overrightarrow{n}$, or vice versa. This version gives good performance (details below), but has the disadvantage of requiring many labelled examples of phrases $an$. They therefore present a second version, in which a general "conceptual metaphor" mapping $C_M$ for adjectives is learned, to minimise the error between $\overrightarrow{an}$ and the product $C_M A_{L(a)} \overrightarrow{n}$ for metaphoric examples. This reduces performance, but does not require labelled examples for every adjective $a$. However, it does still require a large number of observations of every phrase $an$ in context, to build the vector $\overrightarrow{an}$; and is therefore not suitable for analysing novel phrases as yet unseen.

**Divergent views** Our own approach, while it is situated within the distributional semantic paradigm, is inspired by theoretical considerations which diverge somewhat from the doctrinaire stance on conceptual metaphors and selectional violation. In their *deflationary account of metaphors*, Wilson and Sperber (2012) argue that metaphor is properly understood as a particularly

salient instance of the pragmatically resolved underspecification inherent in all language use. Similarly, Carston (2012) suggests that metaphor – and indeed all language use – involves the construction of *ad hoc* conceptual schemes in which the implications of an utterance are resolved contextually based on expectations of maximal relevance. Davidson (1978) has even gone so far as to suggest that metaphor should always primarily be taken literally in the first instance, a controversial view which has received more recent support from Reimer (2001), who engages with the Davidsonian rejection of cognitive content in order to tease out a distinction between the lexical meaning of words, which exist in an encyclopedic sense, versus the pragmatic use of language.

These theoretical considerations, coupled with psycholinguistic research indicating a neurological ambiguity between the processing of ostensibly literal versus metaphoric language (McElree and Nordlie, 1999; Wolff and Gentner, 2011), motivates us to propose a model for metaphor identification which uses information about input words to project an *ad hoc* geometry, in which words and their associations are mapped to features of a contextually specific space. A distributional semantic model of this kind has been proposed by McGregor et al. (2015) for concept discovery, and Agres et al. (2016) have suggested its appropriateness for modelling metaphoricity; we take this as our starting point (more details below).

## 3   A Method for Contextual Geometry

We seek to redress the evident schism between pragmatic and computational accounts of metaphor by taking into account the significance of context in the online construction of situated conceptualisations. With this in mind, we characterise our model in terms of three desiderata:

1. Spaces of word representations for identifying metaphoricity should be in some sense selected contextually;

2. The generation of these spaces should be independent of *a priori* ratings of objective metaphoricity or literalness;

3. The contextualisation should generate a space which is geometrically interpretable.

Following McGregor et al. (2015), the model is based on a standard distributional vector space,

using pointwise mutual information (PMI) co-occurrence statistics calculated over a large-scale corpus, in this case the December 2014 dump of English language Wikipedia. The top 200,000 most frequent words are taken as the model's vocabulary, and every single word type which appears within a context window of $n$ words on either side of one of these target words is a co-occurrence basis dimension. We use a version of positive PMI as shown in (1), where $n_{w,c}$ is the frequency at which target word $w$ occurs in the context of context word $c$; $n_w$ is the total frequency of $w$; $n_c$ is the total frequency at which $c$ occurs as a context word; $W$ is the total count of vocabulary word tokens in the corpus, and $a$ is a smoothing constant (set in this case at 10,000):

$$PMI_{w,c} = \log_2 \left( \frac{n_{w,c} \times W}{n_w \times (n_c + a)} + 1 \right) \quad (1)$$

Note that rather than discarding negative PMI values as in standard approaches, they are weighted by the $+1$ term, ensuring that all values will be positive, with a value of 0 indicating that $w$ and $c$ are never observed to co-occur. With a total of c.7.5 million word types in the corpus, this results in a very large, very sparse matrix of 200,000 rows representing vocabulary word-vectors.

This matrix can now serve as a base space which can be contextualised by selecting relevant lower-dimensional subspaces based on an analysis of input words. McGregor et al. (2015) use this to discover ad-hoc concepts; here, potentially metaphoric adjective-noun word pairs will be used as input, and dimensions selected based on the independent and correlated properties of the word-vectors for each word in the pair. As these dimensions relate to PMI between target and context words, they contain information about a word's most salient associations. We expect the combination of these properties to be informative about the conceptual relationship denoted by the phrase in question; and therefore predict that the geometric relationships of the words within this subspace will be indicative of the likelihood of an assessment of metaphoric versus literal usage.

With our base space established, the primary parameter of our experiment will therefore be the criteria for selecting a subspace based on any given word pair. Following and expanding upon Agres et al. (2016), we experiment with five different techniques for selecting $k$-dimensional subspaces

based on an adjective-noun word pair $\{a, n\}$ and corresponding word vectors $\overrightarrow{a}$ and $\overrightarrow{n}$:

**Joint** Select all dimensions with non-zero PMI values for both $\overrightarrow{a}$ and $\overrightarrow{n}$; then select the top $k$ dimensions with the highest mean value across $\overrightarrow{a}$ and $\overrightarrow{n}$ from this subset.

**Independent** Select the top $k/2$ dimensions for $\overrightarrow{a}$ and $\overrightarrow{n}$ independently.

**Zipped** Select all mutually non-zero dimensions; then select the top dimensions for $\overrightarrow{a}$ and $\overrightarrow{n}$ independently from the subset.

**Adjective Only** Select the top $k$ dimensions for $\overrightarrow{a}$, regardless of their values for $n$.

**Noun Only** Select the top $k$ dimensions for $\overrightarrow{n}$.

In each of these subspaces, we consider the situation of $\overrightarrow{a}$ and $\overrightarrow{n}$ in terms of a set of triangulations between these vectors, the origin of the space, and a line extending from the origin through the centre of the space, as illustrated in Figure 1. The points A and B correspond to the word-vectors for the adjective and the noun in a potentially metaphoric adjective-noun phrase. The points of the triangle CDE sit on the surface of a hypersphere emanating from the origin of the subspace, with C and D corresponding to the vectors of A and B normalised to length $r$ respectively. The point E corresponds to the likewise normalised value of a vector composed of the mean PMI values of each all points on each dimension delineating the subspace—so, a point along a scaled centre-line, effectively.

We predict that ratings of metaphoricity will correlate with features of the geometry of these triangulations in subspaces. In particular, we predict that, in the case of spaces generated from adjective-noun pairs that would be considered typically metaphoric, sides AB and CD will tend to increase in length, with the points A and B tending to diverge on a dimension-by-dimension basis as the contextual subspace selected by the corresponding input terms and word-vectors becomes less consistent. We likewise expect points A and B to move closer to the origin in the cases of jointly selected subspaces, or, in the cases of the singly selected subspaces, the point corresponding to the word-vector not involved in the selection will recede. In other words, we predict that the geometry of the space will reflect a dithering of correspondence in what might be considered the *way of talking about* each of the things denoted by the input

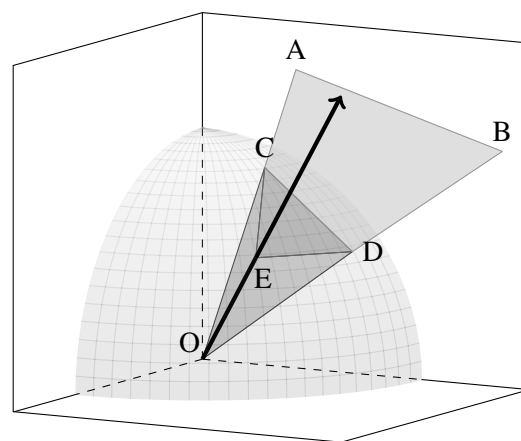

Figure 1: A Metaphor In Space: Word-vectors for adjective-noun pairs are projected into contextualised subspaces where metaphoricity is found to correlate with geometrical relationships between the word-vectors and the centre of their subspace.

terms as the conceptualisation suggested by their composition moves towards what is perceived as a more metaphoric interpretation of the language.

## 4 Experiment and Results

In order to test the contextual, geometric model described in Section 3, we apply the method outlined above to the dataset of Gutiérrez et al. (2016). The data consists of 8,592 adjective-noun dyads, each using one of 23 adjectives chosen for their propensity to be used in both literal and metaphoric contexts, and each occurring at least ten times in a large-scale composite corpus. Each dyad (adjective-noun pair) is tagged with a binary annotation indicating either metaphoricity or literalness; 53.5% were deemed metaphoric, with each adjective type occurring at an average rate of 0.664 as either metaphoric or literal.

For each dyad, we use each of the techniques described in Section 3 above to project the word-vectors for the adjective and the noun into a contextualised subspace for that adjective-noun pair (i.e. its basis dimensions are the words with contextually salient associations). We then take the lengths of the sides of the triangles illustrated in Figure 1 – triangles ABO and CDE – as features in a logistic regression model, trained to predict the metaphoric/literal label. We use `scikit-learn`'s LogisticRegression class, applying mean-zero standard-deviation-1 normalisation to the features, and evaluate using standard shuffled 10-fold cross-validation (again following

Gutiérrez et al. (2016)). Our evaluation metrics are the standard (micro-)average precision, recall, f-score and accuracy across the 10 folds. Note that while we train the regression model using labelled instances, our input features are only information about the individual words - there is no requirement for, or learning of, phrase representations (thus allowing application to unseen phrases).

We test various versions of our model, experimenting with the five different subspace selection techniques described in Section 3. For each technique, we experiment with varying the number of dimensions used to build the subspaces (we test the top 20, 50, 200, and 500 dimensions); and with the size of the context window used to build the base co-occurrence model (we test 2-word and 5-word windows, i.e. taking context words up to 2 words and 5 words on either side of a target word). F-scores returned for these various parameters are listed in Table 1.

Table 1: F-scores for subspaces of dimensionalities 20, 50, 200, and 500 built from 2+2 and 5+5 word context window models.

|  |  | Joint | Indy | Zip | Adj | Noun |
|---|---|---|---|---|---|---|
| 2+2 | 20 | 0.761 | 0.776 | 0.778 | 0.727 | 0.750 |
|  | 50 | 0.776 | 0.784 | 0.790 | 0.727 | 0.750 |
|  | 200 | 0.795 | 0.793 | 0.795 | 0.755 | 0.761 |
|  | 500 | 0.790 | 0.790 | 0.784 | 0.757 | 0.765 |
| 5+5 | 20 | 0.772 | 0.782 | 0.785 | 0.751 | 0.757 |
|  | 50 | 0.786 | 0.798 | 0.798 | 0.756 | 0.764 |
|  | 200 | 0.806 | 0.802 | 0.813 | 0.783 | 0.771 |
|  | 500 | 0.811 | 0.806 | 0.812 | 0.774 | 0.777 |

We compare to the method of Gutiérrez et al. (2016), which learns tensors for each adjective; as described above, they give two versions, one fully supervised version which learns separate tensors for literal and metaphoric senses from tagged data, and one which uses only the literal-sense tensors for each adjective and combines these with a tensor corresponding to the mapping from literal to metaphoric senses in general. They take the second method as more suitable for "unseen" use cases, although it still requires enough observations of the adjective-noun pair to build its own phrase vector. As our method uses no information about the whole phrase occurrence, and is thus suited for truly unseen cases, a more relevant comparison is the second version.

However, while they evaluate their first version ("V1" in our tables below) on the full dataset, obtaining an f-score of 0.817, they only evaluate the second version ("V2") on a reduced subset of the data due to their data and labelling requirements. For this, they obtain an f-score of 0.793, but it seems likely that performance on the full dataset would be lower, as this is the case for V1 (full set f-score 0.817, reduced set 0.838). We give results for comparison in Table 2, but take the V2 results as the more relevant comparison, although they may over-estimate full dataset performance.[1] We also show a majority-class baseline score, and a score for a logistic regression based on cosine similarity between adjectives and nouns as returned by `word2vec` (Mikolov et al., 2013).

Table 2: Comparison between results reported by Gutiérrez et al. (2016) (G2016 V1/V2), our methodology, word2vec cosine similarities, and a majority-class (all-metaphoric) baseline.

|  | prec | rec | f-score | acc |
|---|---|---|---|---|
| GEOMETRY | 0.791 | 0.836 | 0.813 | 0.794 |
| G2016 V1 | 0.842 | 0.793 | 0.817 | 0.809 |
| G2016 V2 | 0.716 | 0.819 | 0.793 | 0.804 |
| WORD2VEC | 0.580 | 0.737 | 0.649 | 0.574 |
| BASELINE | 0.535 | 1.000 | 0.697 | 0.535 |

## 5 Analysis

First of all, our model gives f-scores close to Gutiérrez et al.'s V1 model, and outperforms the more relevant V2 model (both significant at $p<0.05$ using $\chi^2$). The mechanisms for achieving these results, however, are evidently somewhat different. It is noteworthy – and indeed impressive – that the compositional V1 model gives particularly strong precision scores, given that, with a dataset slightly biased towards the metaphoric, a strategy of simply calling everything a metaphor would result in perfect recall, a relatively high f-score, but low precision (see the baseline results).

One of the strengths of our model is that it does not require pre-built phrase vectors based on observations across a corpus; we could, in principle, evaluate the metaphoricity of any given com-

---

[1] Evaluation on the reduced set is not currently possible, as the identity of the subset used is not available. Note that Table 2 gives the V1 and V2 results as reported in the original paper, although the V2 combination of recall, precision and f-score does not quite add up; precision should perhaps be higher, or f-score lower.

position of words within our vocabulary by projecting a contextual subspace based on an analysis of all the independent words in the phrase. Gutiérrez et al. acknowledge the desirability of such a model, although both their V1 and V2 methods require the construction of and comparison with phrase vectors. Their V2 method does not learn individual metaphoric/literal tensors for adjectives, so is more general: but returns slightly weaker results (f = 0.793 on a reduced dataset) than the original, phrase-dependent methodology (f = 0.838), with the balance tipped more towards recall than precision, as with our model. As we're not aware of the phrases which were included in the reduced dataset, however, a direct comparison isn't possible at this point.

Returning to the parametric comparison of results presented in Table 1, the progression of f-scores for each technique for subspace progression bears further analysis. The first thing to note is that the models based on larger co-occurrence windows do better, a finding which lines up with Sahlgren's (2008) description of the tendency towards the semantic (*syntagmatic*) and away from the syntactic (*paradigmatic*) as this co-occurrence window widens. This finding could motivate an exploration of spaces employing broader co-occurrence windows, not to mention spaces spaces populated by word-vectors consisting of more sequentially and syntactically nuanced features.

Next, clearly, and not surprisingly, the spaces that take information from both terms involved in the candidate phrase perform better than those that consider only one of the two words, and it must be noted that the best-performing *joint* and *zipped* subspaces will converge as the dimensionality parameter increases, with the subspace projected eventually including all the dimensions with non-zero values for both input words in all three cases, while the *independent* subspaces will eventually contain all dimensions with non-zero values for either input word. It's likewise noteworthy that the zipped technique, which returns subspaces consisting of dimensions with relatively independent saliency for each input term, does best at dimensionalities of up to 200, whereas both the joint and independent spaces increase in performance towards 500 dimensions. We interpret this as indicating that the dimensions of a subspace most predictive of metaphor contain some combination of information about the terms involved in the candidate phrase that is neither maximally salient to both (or all) terms nor particularly skewed towards one term. Again, a more nuanced approach to subspace projection is probably available and warrants further research.

|  | Joint | Indy | Zip | Adj | Noun |
|---|---|---|---|---|---|
| AO | 0.669 | 0.616 | 0.660 | 0.632 | 0.721 |
| BO | 0.739 | 0.747 | 0.736 | 0.753 | 0.734 |
| AB | 0.644 | 0.657 | 0.638 | 0.617 | 0.700 |
| CE | 0.652 | 0.691 | 0.658 | 0.657 | 0.648 |
| DE | 0.651 | 0.702 | 0.668 | 0.629 | 0.690 |
| CD | 0.651 | 0.689 | 0.679 | 0.628 | 0.646 |
| ABO | 0.794 | 0.784 | 0.801 | 0.767 | 0.766 |
| ADE | 0.682 | 0.701 | 0.696 | 0.680 | 0.650 |

Table 3: F-scores for metaphoricity classification based on independent measures of sides of triangles and on two separate triangles, taking point A in Figure 1 to be correspond to the adjective word-vector and point B to correspond to the noun.

In order to better understand the geometry selected by our analysis, we now perform logistic regressions on the two triangles delineated in our subspaces as well as each of the features of the geometry – each side of both triangles – independently, and likewise return regression results for the sides of each of the two (ABO and CDE in Figure 1) separately. The results are reported in Table 3. Notably, the length of side BO, corresponding to the norm of the noun word-vector, is by far the feature most predictive of metaphor. That the norm of the adjective word-vector (side AO) is not nearly as predictive is to be expected given the nature of the dataset: as the data was constructed specifically to test adjectives that tend to occur in both metaphoric and literal adjective-noun compositions, each adjective contributes to the construction of both metaphoric and literal subspaces. With this in mind, it also makes sense that AO is relatively predicative in the noun-only type subspace, where the adjectival word-vector isn't contributing at all to the selection of dimensions.

On the other hand, it's also notable that AO is relatively predictive in other subspaces (*relatively* is itself relative here—aside from the noun norm BO, very few of the features independently outperform the majority class baseline of $f = 0.697$). We interpret this to mean that, notwithstanding the legitimate efforts on the part of the dataset designers to choose versatile adjectives, there is

a general tendency for some adjectives to be involved in more metaphoric compositions than others. As mentioned in Section 4, the typical balance of adjectives in this dataset is an almost 2:1 preference towards selecting either metaphoric or literal compositions. The strategy of simply determining, based on observations across the annotated data, that a given adjective was always either metaphoric or literal would therefore result in both an accuracy rate and an f-score of 0.664.

Finally, the relatively poor predictiveness of triangle ADE – the lengths between the normalised input word-vectors and a normlised centre line – bears further analysis, particularly in light of the fairly even distribution of performances between geometric features other than BO. In order to understand how each the shape of these related triangles shifts as we move from the metaphoric to the literal end of the spectrum of adjective-noun compositions, we examine the geometry of specific cases of metaphors in space, as visualised in Figure 2. Treating our logistic regression as a generative model, we extract the word-pair deemed to be most metaphoric in 200-dimensional zipped-type spaces, and similarly the word-pair considered to be least metaphoric and a word-pair that falls close to the centre of the model's predictions. As can be seen, there is an lengthening of the distance between adjective and noun moving from the metaphoric to the neutral example, and then an extension of the disance of both points from the origin moving from the neutral to the literal instance.

We were admittedly initially surprised by this evident bulging at the middle of the metaphoric-literal spectrum; we had predicted a steady widening and corresponding move towards the periphery of subspaces as word pairs move towards the more metaphoric. In order to check if this trend is general rather than just a peculiarity of the metaphors examined in Figure 2, we calculate the average ratio of the length of AB to the distance of its midpoint from the origin for the 500 most metaphoric, most neutral, and most literal adjective-noun pairs in the dataset based on our regression. Results are reported in Table 4, considering not only the zipped type subspace anecdotally depicted in Figure 2 but also the two other subspace varieties selected involving information from both input word-vectors. (The subspaces selected based only on an analysis of either noun or adjective vectors tend to be characterised by significant difference

|  | metaphoric | neutral | literal |
|---|---|---|---|
| JOINT | 0.767 | 0.870 | 0.792 |
| INDY | 1.246 | 1.312 | 1.306 |
| ZIPPED | 0.923 | 1.028 | 0.977 |

Table 4: Average ratio of distance between adjective and noun word-vectors over mean norm of those vectors, for different regions of the metaphoric-literal spectrum and different subspace selection techniques.

between norms of each projected vector, so this analysis is less meaningful there.)

Significantly, the trend is evident across all three multi-word input subspaces, with especially pronounced differences in the joint type subspaces. It would seem that the move from metaphoric to literal is in fact characterised by a two part expansion, beginning with a widening of the space between the input terms as they become more neutrally (or perhaps ambiguously) interpreted and then an extension away from the origin as they become more literally interpreted. What seems to be happening here is actually an increase in the dimensions available to the model as the terms move towards the neutral, but at the same time a diminishing of the correspondence between the input words and so a widening of the space between them on a dimension-by-dimension basis. The end result is actually a kind of horseshoe phenomenon, with both literal and metaphoric compositions tending towards one another in the subspaces they select, but with literal compositions moving away from the origin as there tends to be more correspondence between the set of salient (ie, high valued) co-occurrence dimensions for each word. Meanwhile, what we now refer to as *ambiguous* rather than neutral compositions tend to select for subspaces where a wider range of dimensions have some information about both terms, but there tends to be less correspondence between each word across these dimensions.

Returning to some of the theoretical suppositions outlined in Section 2, we recall that we postulated that metaphor, and indeed language in general, is always ultimately determined contextually. We outline a small set of examples sampled from the middle and either end of the spectrum outlined by our regression to illustrate this point:

metaphoric: *murky franc, clear result, brilliant exception, smooth solicitor, icy formality*

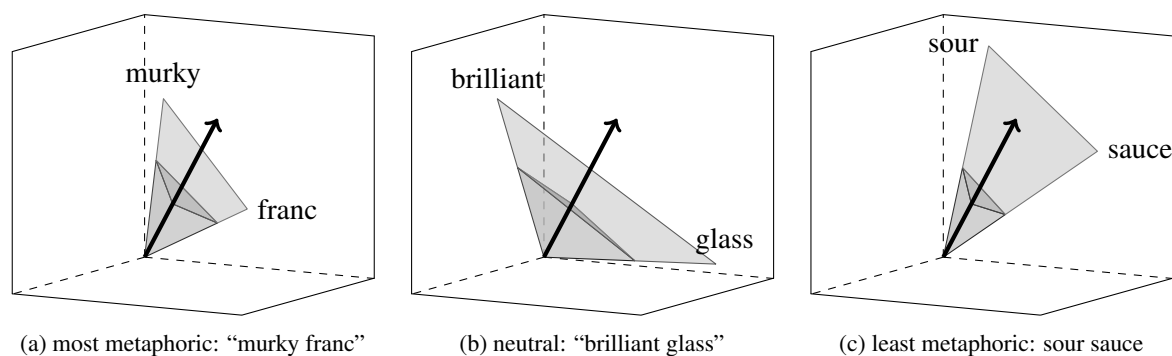

(a) most metaphoric: "murky franc" (b) neutral: "brilliant glass" (c) least metaphoric: sour sauce

Figure 2: Phrases Across the Spectrum of Metaphoricity: Three phrases, the most metaphoric, least metaphoric, and one sampled from the middle of the spectrum, based on an analysis of subspaces projected by our model. This projection preserves all lengths within the triangles, as well as the position of the smaller triangle tangentialed to a hypersphere eminating from the origin.

ambiguous: *brilliant glass, bitter spirit, dense information, shallow center, heavy betting*

literal: *sour sauce, sweet garlic, sour cabbage, sour lemon, sweet sauce*

We note in particular that phrases such as "bitter spirit" and "shallow center" are open to interpretation depending on the context that they are used. Indeed, we maintain that all these phrases are, to some extent, open to interpretation when taken out of context: a "smooth solicitor" could conceivably be a soft-skinned lawyer, while "sweet sauce" might describe a sauce which is generally desirable rather than particularly sugary.

So here at last we return to the issue of metaphor in context, and make two final comments by way of reinforcing observations which have been made frequently in the theoretical literature. The first is that metaphor is probably properly construed as being an extent of a spectrum, and a spectrum that is not clearly defined in absolute terms but rather is probably specific to a contextual expectation. The second point follows on from this: metaphor is contextual, and and given adjective-noun phrase could probably, at a stretch, be forced into either a metaphoric or literal interpretation under the correct conditions. So, while our model provides the basic equipment for an unsupervised analysis on any given word pair, we feel it also lays the groundwork for generating more contextually nuanced spaces in which conceptual relationships map to the geometric situations of word-vectors.

## 6 Conclusion

The conceptual metaphor model — with metaphors as mappings between conceptual schemas — has been rightly influential in computational linguistics; it is not our intention to be iconoclastic. However, our results suggest that it might be enhanced by combining it with the insight that lexical semantics is always resolved contextually. Conceptual schemas can be recast as geometries within ad-hoc, contextualised subspaces, delineated by dimensions that are tractable, interpretable features of language use which relate to salient co-occurrence associations; this view, even simply implemented as here, gives accuracy comparable to state-of-the-art approaches while retaining the *zero-shot* ability to apply to truly unseen examples.

The primary features of our method are its contextual dynamism and its geometry. Through the online selection of subspaces based on context-specific input, we're able to build *ad hoc* subspaces in which novel conceptual relationships emerge as geometric features. A strength of this model is its simplicity, and likewise its availability as an interpretable framework that can be easily analysed using standard classification and regression techniques. In principle, any words in our vocabulary could be projected into a subspace, and we might expect the geometry of such subspaces to do more work in the long run, issuing features involving other semantic relationships and compositionality. Ultimately, we hold that language itself essentially affords conceptualisation: while words are not ideas and thought are not words, language provides a structure with which to, as Barsalou (1993) puts it, *haphazardly* compose novel concepts in an unpredictable world. Here we have attempted to consider how a computational approach to such a flexible model might work.

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
