# Peer review of "A Geometric Contextual Model for Identifying Unseen Metaphors"

_ACL 2017 — decision unknown_

[Official Review · Reviewer 1 · rating 3 · confidence 4]
soundness 3 · originality 4 · clarity 4 · impact 3 · substance 3 · appropriateness 5 · meaningful comparison 4 · presentation format Poster

This paper proposes an approach for classifying literal and metaphoric
adjective-noun pairs. The authors create a word-context matrix for adjectives
and nouns where each element of the matrix is the PMI score. They then use
different methods for selecting dimensions of this matrix to represent each
noun/adjective as a vector. The geometric properties of average, nouns, and
adjective vectors and their normalized versions are used as features in
training a regression model for classifying the pairs to literal or metaphor
expressions. Their approach performs similarly to previous work that learns a
vector representation for each adjective.

Supervision and zero-shot learning. The authors argue that their approach
requires less supervision (compared to previous work)  and can do zero-shot
learning. I don’t think this is quite right and given that it seems to be one
of the main points of the paper, I think it is worth clarifying. The approach
proposed in the paper is a supervised classification task: The authors form
vector representations from co-occurrence statistics, and then use the
properties of these representations and the gold-standard labels of each pair
to train a classifier. The model (similarly to any other supervised classifier)
can be tested on words that did not occur in the training data; but, the model
does not learn from such examples. Moreover, those words are not really
“unseen” because the model needs to have a vector representation of those
words.

Interpretation of the results. The authors provide a good overview of the
previous related work on metaphors. However, I am not sure what the intuition
about their approach is (that is, using the geometric properties such as vector
length in identifying metaphors). For example, why are the normalized vectors
considered? It seems that they don’t contribute to a better performance.
Moreover, the most predictive feature is the noun vector; the authors explain
that this is a side effect of the data which is collected such that each
adjective occurs in both metaphoric and literal expressions. (As a result, the
adjective vector is less predictive.) It seems that the proposed approach might
be only suitable for the given data. This shortcoming is two-fold: (a) From the
theoretical perspective (and especially since the paper is submitted to the
cognitive track), it is not clear what we learn about theories of metaphor
processing. (b) From the NLP applications standpoint, I am not sure how
generalizable this approach is compared to the compositional models.

Novelty. The proposed approach for representing noun/adjective vectors is very
similar to that of Agres et al. It seems that the main contribution of the
paper is that they use the geometric properties to classify the vectors.

[Official Review · Reviewer 2 · rating 3 · confidence 4]
soundness 3 · originality 4 · clarity 4 · impact 3 · substance 3 · appropriateness 5 · meaningful comparison 4 · presentation format Poster

This paper presents a  method for metaphor identification based on geometric
approach. Certainly, very interesting piece of work. I enjoyed learning a
completely new perspective. However, I have a few issues, I like them to be
addressed by the authors. I would like to read author's response on the
following issues.

- Strengths:

* A geometric approach to metaphor interpretation is a new research strand
altogether. 
* The paper is well written.
* Author's claim is the beauty of their model lies in its simplicity, I do
agree with their claim. But the implication of the simplicity is not been
addressed in simple ways. Please refer the weakness section.

- Weaknesses:
Regarding writing
===============
No doubt the paper is well-written. But the major issue with the paper is its
lucidness. Indeed, poetic language, elegance is applaud-able, but clarity in
scientific writing is very much needed. 
I hope you will agree with most of the stuff being articulated here:
https://chairs-blog.acl2017.org/2017/02/02/last-minute-writing-advice/

Let me put my objections on writing here:
* "while providing a method which is effectively zero-shot"..left readers in
the blank. The notion of zero-shot has not been introduced yet!
* Figure 2: most, neutral, least - metaphoric. How did you arrive at such
differentiations?
* Talk more about data. Otherwise, the method is less intuitive.
* I enjoyed reading the analysis section. But it is not clear why the proposed
simple (as claimed) method can over-perform than other existing techniques?
Putting some examples would be better, I believe.

Technicality
============
 "A strength of this model is its simplicity" - indeed, but the implication is
not vivid from the writing. Mathematical and technical definition of a problem
is one aspect, but the implication from the definition is quite hard to be
understood. When that's the note-able contribution of the paper. Comparing to
previous research this paper shows only marginal accuracy gain.

* Comparison only with one previous work and then claiming that the method is
capable of zero-shot, is slightly overstated. Is the method extendable to
Twitter, let's say.

- General Discussion: